# Put in a “Ca^2+^ll” to Acute Myeloid Leukemia

**DOI:** 10.3390/cells11030543

**Published:** 2022-02-04

**Authors:** Clara Lewuillon, Marie-Océane Laguillaumie, Bruno Quesnel, Thierry Idziorek, Yasmine Touil, Loïc Lemonnier

**Affiliations:** 1Univ. Lille, CNRS, Inserm, CHU Lille, UMR9020-U1277—CANTHER—Cancer Heterogeneity Plasticity and Resistance to Therapies, F-59000 Lille, France; clara.lewuillon@inserm.fr (C.L.); marie-oceane.laguillaumie@inserm.fr (M.-O.L.); bruno.quesnel@chru-lille.fr (B.Q.); thierry.idziorek@inserm.fr (T.I.); yasmine.touil@inserm.fr (Y.T.); 2Univ. Lille, Inserm, U1003—PHYCEL—Physiologie Cellulaire, F-59000 Lille, France; 3Laboratory of Excellence, Ion Channels Science and Therapeutics, F-59655 Villeneuve d’Ascq, France

**Keywords:** acute myeloid leukemia, calcium signaling, leukemic stem cells, cell metabolism, microenvironment, chemotherapies

## Abstract

Acute myeloid leukemia (AML) is a clonal disorder characterized by genetic aberrations in myeloid primitive cells (blasts) which lead to their defective maturation/function and their proliferation in the bone marrow (BM) and blood of affected individuals. Current intensive chemotherapy protocols result in complete remission in 50% to 80% of AML patients depending on their age and the AML type involved. While alterations in calcium signaling have been extensively studied in solid tumors, little is known about the role of calcium in most hematologic malignancies, including AML. Our purpose with this review is to raise awareness about this issue and to present (i) the role of calcium signaling in AML cell proliferation and differentiation and in the quiescence of hematopoietic stem cells; (ii) the interplay between mitochondria, metabolism, and oxidative stress; (iii) the effect of the BM microenvironment on AML cell fate; and finally (iv) the mechanism by which chemotherapeutic treatments modify calcium homeostasis in AML cells.

## 1. Introduction

Acute myeloid leukemia (AML) is a heterogeneous hematologic malignancy at the biological, molecular, and clinical levels. AML is characterized by clonal amplification and the loss of differentiation of myeloid precursors (blasts) in the bone marrow (BM) and peripheral blood. AML is a pathology of aged adults, with more than 50% of patients identified at age > 65 years. Despite a tremendous effort to decipher genetic aberrations such as leukemia-associated chromosomal translocations and inversions, as well as multiple somatically acquired mutations that affect genes of different functions that improve prognosis, the AML patient survival rate remains poor and is largely age-dependent [1]. AML originates from BM hematopoietic stem cells (HSCs). Primary commonly acquired mutations arise in genes involved in the epigenomic process, such as DNMT3A, ASXL1, TET2, IDH1, and IDH2, and already exist in the founding clone. In contrast, secondary mutations involving NPM1 or signaling molecules (e.g., FLT3, RAS gene family) typically occur later during leukemogenesis. In addition, AML can be subdivided into distinct classes, including acute promyelocytic leukemia (APL) and myelodysplastic-syndrome-related AML, which involve different genetic aberrations (Myc, etc.) [2].

Calcium ions are the main second messengers in cells and are crucial for cell fate and survival. Intracellular Ca^2+^ homeostasis relies on the organized activity of various Ca^2+^ channels, pumps, and exchangers, which maintain a precise Ca^2+^ concentration in the cytoplasm and organelles such as the endoplasmic reticulum (ER), mitochondria, lysosomes, and nucleus [3,4,5]. Transient or sustained activation of this machinery regulates changes in the duration and levels of intracellular calcium, thereby governing calcium-dependent physiological processes.

Alterations in calcium signaling and homeostasis affect numerous cellular functions and are involved in various pathological states, including cancer. Dysregulated calcium signaling cascades have been shown to result in processes crucial for uncontrolled proliferation and tumorigenesis. These processes include gene transcription, regulation of the cell cycle, proliferation, metabolism, apoptosis, autophagy, and cell migration and may affect the development of resistance to cancer therapies [6].

Alterations in calcium signaling have been extensively studied in solid cancers. There is much less knowledge of hematologic malignancies such as myeloid leukemia, except for chronic myeloid leukemia (CML), for which the tyrosine kinase activity of Bcr–Abl seems to regulate calcium homeostasis [7].

Two recent publications highlight the important role of calcium in the maintenance of normal stem cells, both active and dormant. Luchsinger et al. show that a low-calcium medium increases the viability of HSCs in vitro [8]. These have a low cytoplasmic calcium concentration that is maintained by glycolytic activity. Fukushima et al. developed another strategy by which a non-phosphorylatable fluorescent marker allows in vivo discrimination between dormant and active stem cells [9]. It shows the implication of calcium concentration in dormancy and that marrow reconstitution by HSC is favored by a high concentration of cytoplasmic calcium. The bone matrix allows the localization of hematopoietic stem cells in niches that are critical for their regulation [10]. In bone, these niches are mainly located in vascularized areas and more rarely in trabecular areas. The latter, however, have a strong potential for self-renewal and reconstruction of hematopoiesis that emphasizes their importance, especially for the development of leukemic cells [11]. There is a real interest in studying the precise role of calcium signaling in AML, which remains mostly unclear.

Taking into account the current knowledge of calcium signaling in cancer cells as well as in normal hematopoiesis, our purpose is to shed light on four peculiar topics: (i) the quiescence of these hematopoietic stem cells and AML cell proliferation and differentiation; (ii) the role of mitochondria, metabolism, and oxidative stress; (iii) the effect of the BM microenvironment on AML cells; and finally (iv) therapeutic approaches.

## 2. The Role of Calcium Homeostasis in AML Cell Proliferation and Differentiation

The dysregulation of calcium signaling and homeostasis impacts numerous cellular functions. Several Ca^2+^-dependent signaling pathways are therefore involved in cancer initiation and development. The precise control of intracellular calcium concentration is crucial for the modulation of many signaling pathways and Ca^2+^-regulated proteins involved in specific cellular processes, including the regulation of key cellular functions such as the cell cycle, proliferation, and differentiation.

Our review focuses on the role of Ca^2+^ channels and downstream intracellular signaling pathways in regulating proliferation and differentiation in acute myeloid cell lines or primary AML samples. Numerous reviews have presented the remodeling of intracellular Ca^2+^ homeostasis in cancer cells from solid tumors but very few in hematologic malignancies and none in acute myeloid leukemia. We here focus on studies that highlight a functional role for Ca^2+^ channels and downstream intracellular signaling pathways that lead to changes in proliferation and differentiation in acute myeloid leukemia. Our review also focuses on the regulation of quiescence by calcium signaling in normal hematopoietic stem cells and leukemic stem cells.

### 2.1. Calcium and Cell Cycle Regulation in AML

Cell division and proliferation are ruled by the cell cycle, which is a four-stage process: the first gap phase G1; the S phase, in which DNA replication occurs; the second gap phase G2; and the M phase, or mitosis, in which the DNA and cytoplasmic material are shared between two new daughter cells. The progression between these different phases is tightly regulated, and variations in intracellular Ca^2+^ [Ca^2+^]_i_ play a key role throughout the cell cycle, namely during the early G1 phase and at the G1/S and G2/M transitions.

The initiation of centrosomal duplication at the G1/S phase is also dependent on Ca^2+^ and calmodulin (CaM) [12,13]. CaM appears to be mobilized early after mitogenic stimulation and late in the G1 phase near the G1/S transition [14,15]. In late G1, Ca^2+^/CaM is required before the restriction point and pRb phosphorylation [16]. In AML, it has been shown that Ca^2+^ and CaM are involved in regulating cell proliferation in HL-60 human promyelocytic leukemia cells. Indeed, cytosolic calmodulin levels were increased according to the cell cycle phase and more generally during cell cycle progression. Additionally, it has been shown that calmodulin antagonists slow cell growth in a concentration-dependent manner [17], thus underlying the critical role of the Ca^2+^/CaM pathway in leukemia cells.

CaM kinase II (CaMKII) is necessary for cell cycle progression [18]. Inhibition of CaMK activity was shown to inhibit cell proliferation and is correlated with growth arrest in AML. Monaco et al. reported that inhibition of CaMKII activity results in an upregulation of CaMKIV mRNA and protein expression in leukemia cell lines. Interestingly, AML cells (primary cells and cell lines) expressing CaMKIV show elevated levels of Cdk inhibitors p27 (kip1) and p16 (ink4a) and reduced levels of cyclins A, B1, and D1. These findings indicated potential cross-talk between CaMKII and CaMKIV and suggest that CaMKII could suppress the expression of CaMKIV to promote leukemia cell proliferation [19]. Nevertheless, another study showed contradictory results regarding the effect of CAMKs on AML proliferation. It has been shown that the suppression of CaMKs (CAMKIV) coupled with human leukocyte immunoglobulin-like receptor B2 (LILRB2) signaling is associated with decreased human acute leukemia proliferation in vitro and in vivo [20].

Calcineurin, a calcium-dependent phosphatase, also plays a crucial role in the transition through the G1 and S phases and was shown to regulate the expression of cyclins A and E [21] and the accumulation of cyclin D1 [18]. Calcineurin is also very well known to activate NFAT and to mobilize MYC [22,23] in regulating cyclins E and E2F, which brings an additional link between calcium-dependent pathways and proliferation. However, the role of calcineurin in AML and its precise function in the regulation of cell cycle checkpoints in these malignant hemopathies are unknown. Nonetheless, one study showed that the activity of calcineurin was decreased by 85% in the sera of patients diagnosed with AML, while no significant changes in calmodulin or calcineurin levels were observed [24].

### 2.2. Calcium Channels and Proliferation in AML

The calcium/calcineurin/NFAT pathway is one of the main mechanisms that can be activated by calcium influx through membrane calcium channels. The use of calcium channel inhibitors has strengthened the notion that calcium influx plays a critical role in cell proliferation. The antiproliferative effects of the inhibition of calcium channels have been shown in numerous tissues. In AML cells, the transient receptor potential melastatin 2 (TRPM2) ion channel displayed high expression compared to CD34+ healthy precursor cells, and its suppression inhibited the proliferation of leukemia cells. These findings show that TRPM2 has an important role in AML proliferation mediated by the regulation of key transcription factors, such as ATF4 and CREB [25]. It was also shown that the dextroisomer r-verapamil, which inhibits Cav1.2, an L-type calcium channel, by binding to a specific area of its α-1 subunit, causes dose-dependent inhibition of leukemic cell proliferation. These results were observed in blast cells derived from patients with AML [26].

Calcium transport could be achieved by the activation of upstream partners such as inositol 1,4,5-trisphosphate receptor type 2 (ITPR2), which is located in the endoplasmic reticulum membrane and regulates the mobilization of intracellular Ca^2+^ stores through SERCA pumps and/or exchangers. Thus, ITPR2 plays a pivotal role in intracellular Ca^2+^ signaling and subsequently in the cell cycle and proliferation. ITPR2 expression is increased in AML patients with a normal karyotype compared to healthy patients. In a cohort of 157 AML patients, high ITPR2 expression was associated with dramatically shorter overall survival and event-free survival [27]. Further investigations are needed to determine the precise calcium-dependent mechanism underlying the aggressive phenotype of leukemic cells with high ITPR2 expression in AML.

### 2.3. Implication of Notch and Ca^2+^ Signaling in AML Proliferation

It has been recently shown that Notch increases Ca^2+^ entry by activating calcium-sensing receptors and inhibiting voltage-gated K^+^ channels [26] and that it could also enhance store-operated Ca^2+^ entry [28]. The new human leukemia cell line TMD7, established from blast cells of a patient with de novo acute myeloblastic leukemia, expressed Notch-1 and Notch-2 mRNA. Exposure to recombinant Delta-1 protein, a Notch ligand, significantly increased the proliferation of TMD7 cells [29]. It has also been shown that Notch signaling can maintain the proliferation and survival of the HL60 human promyelocytic leukemia cell line and promotes the phosphorylation of the Rb protein [30]. Other studies have shown, however, that Notch activation can inhibit AML growth and also alter AML-initiating cell compartments [31,32]. The nature of the Notch ligands seems to be crucial in the resulting effect of Notch activation. Indeed, when the ligand used is Jagged1 instead of Delta1, inhibition of proliferation and survival are observed in AML cells. Interestingly, opposite effects of the Notch ligands Jagged1 and Delta1 have been reported on the growth of primary AML cells [33]. While their associated roles in calcium signaling are still under investigation, it would be interesting to clarify the calcium mobilization pattern associated with Notch ligands in AML.

### 2.4. Calcium Involvement in AML Differentiation

Any dysregulation in myeloblast differentiation represents a key cellular and molecular event that could be studied for a better understanding of AML. Complete remission in patients with APL has been achieved using targeted therapies such as all-trans retinoic acid (ATRA) and/or arsenic trioxide [34]. However, the response of non-APL AML patients to treatment remains poor, indicating the need for a better understanding of the differentiation processes in this disease.

Myeloid differentiation involves the activation of new signaling pathways and the acquisition of new effector functions. Cellular calcium homeostasis and calcium-dependent signaling are intimately involved in these processes. Thus, intracellular calcium transport may be significantly remodeled during differentiation [34]. Some studies have shown the link between differentiation and calcium signaling in AML. It has been shown that the modulation of calcium pump expression, specifically of the sarco-endoplasmic reticulum calcium ATPase (SERCA) pumps, is an integral component of the differentiation program of myeloid precursors and indicates that lineage-specific remodeling of the ER occurs during cell maturation. In addition, it was shown that SERCA isoforms may serve as useful markers for the study of myeloid differentiation [34]. In vitro studies have shown the mechanisms implicated in leukemic cell differentiation. The binding of S100A9 to Toll-like receptor 4 (TLR4), which promotes the activation of p38 mitogen-activated protein kinase, extracellular signal-regulated kinases 1 and 2, and Jun N-terminal kinase signaling pathways, could lead to myelomonocytic and monocytic AML cell differentiation [35]. Few studies have been published regarding the implication of calcium in the induction of AML cell differentiation through direct or indirect mechanisms. Calcium ionophores can, by themselves, induce the differentiation of primary human AML cells into dendritic cells [36]. It has also been shown that Ca^2+^ signaling, through the receptor IP3R1, sensitizes cells to the effect of retinoic acid. Differentiation induced by retinoic acid was associated with a significant reduction in c-Myc expression and an increase in membrane tyrosine kinase activity in AML cell lines [37].

### 2.5. Calcium and Cell Cycle Regulation in Normal and Cancer Stem Cells (CSCs)

In recent years, research has started to highlight that calcium channels play a key role in CSC function. Calcium channels are indeed involved in the mechanisms required for their function, leading to cancer progression and treatment resistance. Cancers are most commonly treated with a combination of surgery, chemotherapy, radiotherapy, and immunotherapy. However, several studies have demonstrated that these treatments fail to target or actively select a specific subset of resistant cells termed CSCs [38]. CSCs are characterized by their capacity to remain quiescent and to resist apoptosis, properties generally associated with treatment resistance and tumor relapse [38]. Therefore, deciphering the mechanisms underlying the quiescence of cancer cells or CSCs appears to be crucial in cancer research.

Quiescent cells are found in G0, a resting phase outside of the cell cycle distinct from the G1 phase observed in cycling cells. Regulation of several key players, including cyclins and cyclin-dependent kinases (CDKs), CDK inhibitors, and retinoblastoma protein (Rb), dictate cell fate (G0 entry, cell cycle engagement). Decrease in the expression and activity of cyclin D–CDK4/6 and cyclin E–CDK2 complexes could induce quiescence [39].

The role of calcium in the regulation of stem cell quiescence is still poorly understood. It was shown that the NFATc1 isoform can control hair follicle normal stem cell quiescence by suppressing cyclin-dependent kinase 4 (CDK4) and cell cycle progression [40]. It was also shown that glioblastoma stem-like cells (GSLCs) can be maintained in a quiescent state by decreasing the extracellular pH. Interestingly, in this study, the authors observed that the changes in Ca^2+^ homeostasis appearing during the switch from proliferation to quiescence are determined by store-operated channels (SOCs) since the inhibition of SOCs promotes the quiescence of proliferating GSLCs and induces a dramatic and reversible change in mitochondrial morphology [41].

Regarding the regulation of quiescent HSCs, an increasing number of studies have shown a crucial impact of cytoplasmic calcium concentration [42]. There is a remodeling of intracellular Ca^2+^ homeostasis in HSCs exiting the quiescent state and undergoing commitment. Therefore, Ca^2+^ signaling seems to be highly implicated in hematopoiesis. Nevertheless, studies are currently incomplete and sometimes contradictory. Some studies have shown that Ca^2+^ signals oscillate between keeping HSCs in the quiescent state or activating them when commitment is needed [43]. Toward a differentiation cell fate, it was shown that Ca^2+^ signaling could favor nuclear translocation of the commitment promoter NFAT and degradation of the self-renewal promoter Tet2 [44]. In contrast, an elegant study has shown that quiescent cells display higher intracellular Ca^2+^ concentrations and that niche factors favor [Ca^2+^]_c_ elevation to induce quiescence (possibly via the calmodulin/CaMKs pathway) [9]. Single-cell RNA-seq analyses showed that the gene expression profiles of dormant/quiescent and active HSCs were nearly identical, except for Cdk4/6 activity. Moreover, high-throughput small-molecule screening revealed that high concentrations of cytoplasmic calcium ([Ca^2+^]_c_) were linked to HSC quiescence. These findings indicate that quiescent and active adult HSCs could be distinguished from one another according to the regulation of Cdk4/6 and [Ca^2+^]_c_ [9].

The apparent contradiction of these observations highlights the crucial need to more precisely decipher the mechanisms underlying the implication of Ca^2+^ signaling in the balance between the quiescent and active states of HSCs (Figure 1).

### 2.6. Leukemic Stem Cells (LSCs), Relapse, and Calcium: A Possible Link?

Relapses in AML are a result of the persistence of chemoresistant leukemic cells, also termed minimal residual disease (MRD). Several studies have demonstrated subclonal heterogeneity and a hierarchy of human leukemia-initiating cells (LICs), including LSCs [45], and Bachas et al. showed that a minor subpopulation of LICs responsible for relapse was present at diagnosis in patients [46]. Therefore, a better understanding of the mechanisms of LSC regulation would help to prevent relapse in AML. As described above, some studies have shown that Ca^2+^ signaling is involved in HSCs and in CSC regulation. However, to our knowledge, no study has described the calcium signaling pathways involved in LSC/LIC regulation. Interestingly, RNA-seq analysis of AML patient samples revealed that some NFAT isoforms (NFATc2/3) were overexpressed in AML blasts at relapse compared to blasts at diagnosis [47]. Although the mechanisms were not described in this study, these observations could link calcium-dependent pathways to the properties of leukemic cells causing relapse in AML. Further investigations are needed to better understand the calcium-associated mechanisms underlying relapse in AML, particularly those related to LSCs.

### 2.7. Future Directions

In the above sections, we have highlighted several targets/pathways of interest and hypothetical mechanisms associated with calcium signaling (summarized in Table 1 and in Figure 1) and related to AML proliferation and differentiation or to CSC/HSC quiescence. Nevertheless, few studies have investigated the link between calcium signaling and AML cell differentiation or cell cycle regulation so far, and none were performed on leukemic stem cells. In addition, some observations are contradictory. This may be explained by (i) the different origins of the biological material used, i.e., cells lines vs. primary leukemic cells cultured in different laboratories; (ii) the difficulty in studying rare cells such as HSC or LSC; and (iii) the high heterogeneity of AML disease. Regarding the last point, and as was already mentioned, AML is a complex and highly heterogenous hematologic malignancy which displays a plethora of genetic/epigenetic modifications, possibly impacting calcium signaling and therefore leukemic cell fate toward differentiation and/or proliferation. Interestingly, it was shown in normal myeloid cells that NFAT negatively regulates genes that dictate entry in the cell cycle, such as Cdk4 and Cdk6. This effect was associated with the FLT3 ligand-associated signaling and the phospholipase PLCγ1-dependent calcium influx [48]. This study suggests that NFAT proteins can inhibit proliferative signaling in myeloid cells and interact with the FLT3 receptor, a protein commonly mutated in AML. It could therefore be interesting to focus future efforts to study the impact of Ca^2+^ signaling on key actors of cell division and proliferation in AML cells, depending on the genetic background and/or the differentiation stage of the disease.

Compared to normal myeloid cells, contradictory results regarding the role of NFAT in proliferation have been shown in AML. Indeed, the NFAT isoform c1 was shown to be frequently overexpressed in AML cells bearing the FLT3 mutation and displaying an excessive proliferation [49]. Despite the discordant observations made between normal and leukemic cells, this study underlines the possible link between calcium signaling and cell cycle regulation in AML cells with a specific genetic anomaly background. Further studies are thus still required to better understand the precise role of calcium signaling in AML proliferation and differentiation.

## 3. Mitochondria, Calcium, and AML

One hundred years ago, Warburg proposed that cancer may originate from dysregulated metabolism. Mitochondria are essential organelles that facilitate cell metabolism and energy supply through oxidative phosphorylation (OXPHOS) and acid-driven and/or fatty acid oxidation (FAO) [50,51]; they are also involved in calcium homeostasis and reactive oxygen species production, which play a role in apoptosis (cell death).

The citric acid cycle, also known as the TCA cycle or the Krebs cycle, is a metabolic pathway in all aerobic organisms whose primary function is to oxidize acetyl groups, notably from degraded carbohydrates, fats, and proteins, to restore energy production. The process yields one GTP or ATP and several electrons that circulate through the respiratory chain to allow the formation of additional ATP molecules through OXPHOS. Mitochondrial Ca^2+^ uptake engages aerobic metabolism by triggering the activity of three TCA cycle dehydrogenases: pyruvate dehydrogenase (PDH) via pyruvate dehydrogenase phosphatase 1 (PDP1) and isocitrate and α-ketoglutarate dehydrogenases (IDH and OGDH, respectively) (for review see [52]).

### 3.1. Normal Hematopoiesis

Hematopoiesis takes place in the bone, and HSCs share a specialized microenvironment called a niche with a variety of other cells, such as osteoblasts, osteoclasts, macrophages, adipocytes, and perivascular mesenchymal and endothelial cells, to maintain their pluripotent status. The niche provides hypoxic conditions for the maintenance of HSC quiescence. Quiescent HSCs display low intracellular Ca^2+^ levels in both the cytosol and mitochondria in vivo, which is partly fulfilled via low plasma membrane Ca^2+^ influx activity [43]. Low-calcium environments maintain HSCs’ stem cell features in vitro. In turn, high Ca^2+^ levels increase mitochondrial OXPHOS and compromise stem cell functions while promoting the gene expression of a series of differentiation markers [8], indicating that disturbing mitochondrial Ca^2+^ signaling may reshape the future of HSCs. Fukushima et al. developed a strategy to discriminate quiescent (dormant) from active stem cells using a nonphosphorylatable p27 G0 marker fused to a yellow fluorescent protein, and this strategy bypassed the ER/mitochondrial pathways. Their data show the implication of the calcium concentration in dormancy. Reconstitution of the BM by HSCs in vivo is promoted by a high concentration of cytoplasmic calcium [9], somewhat opposing the results obtained by Luchsinger et al. in their in vitro model.

Under some conditions, such as interferon treatment or 5-fluorouracil-induced BM suppression, HSCs exit the quiescent state and actively enter the cell cycle [43,53]. Umemoto et al. provided evidence that the initiation of cell division starts with increased intracellular Ca^2+^ levels. The resulting enhancement of mitochondrial membrane potential is accompanied by increases in mitochondrial Ca^2+^ levels, mitochondrial superoxide levels, and intracellular ATP content. Inhibiting the increase in intracellular Ca^2+^ via treatment with nifedipine, an antagonist of L-type voltage-gated Ca^2+^ channels (LTCCs), drastically affected mitochondrial Ca^2+^ levels and preserved HSC stem cell features.

One potential mechanism by which mitochondrial Ca^2+^ controls the HSC cell cycle and gene expression is epigenetic regulation. Aside from increased ATP synthesis and ROS production, Ca^2+^ affects the epigenome by allowing the formation of acetyl-CoA and α-ketoglutarate (α-KG). While acetyl-CoA is known to be an essential substrate for histone acetylation, α-KG could also play a role. Lombardi et al. recently provided evidence that the loss of mitochondrial Ca^2+^ uptake stimulates myofibroblast differentiation and fibrosis. They identified that MICU1-mediated MCU triggering elicits a metabolic switch; in this process, α-KG regulates the activity of histone demethylases, including histone lysine demethylases (KDMs) and ten-eleven-translocated (TET) enzymes, and epigenetic remodeling ultimately regulates cell fate [54].

### 3.2. LSCs and AML

Acute myeloid leukemia arises either de novo, in which the earliest mutation triggers the development of the disease, or may expand from other hematological malignancies following the stepwise increase of chromosomal and/or genetic/epigenetic abnormalities [55,56,57,58,59]. Until now, the mechanisms supporting the regulation and clinical significance of the Ca^2+^/mitochondria pathways have remained mostly unidentified in LSCs. LSC metabolism is distinct from that involved in normal HSCs [60,61,62]. While HSCs exploit both OXPHOS and glycolysis, LSCs are defective in glycolysis and mostly rely on amino acid-driven OXPHOS for their basal energy needs but may also oxidize fatty acids to sustain OXPHOS [63].

Fatty acid metabolism has gained substantial interest in hematological malignancies, including AML. LSCs primarily reside in the BM niche in a scarce but adipocyte-rich environment. Tabe et al. and Maher et al. extensively and carefully describe the “ins and outs” of FAO involvement in leukemia [64,65]. In mitochondria, metabolized fatty acids generate NADH and FADH through B-oxidation. These molecules operate as cofactors in the electron transport chain to produce ATP and acetyl-CoA, the latter of which enters the TCA cycle. FAO is essential in AML cells with overexpression of very-long-chain acyl-CoA dehydrogenase (VLCAD) [66] and in AML stem cells that are resistant to the venetoclax/azacitidine regimen [67]. Genetic reduction and pharmacological inhibition of VLCAD impaired mitochondrial respiration and the FAO contribution to the TCA cycle, while pharmacological inhibition of FAO restored sensitivity to venetoclax/azacitidine AML stem cells.

Several proteins involved in mitochondrial calcium signaling present abnormal expression at the plasma membrane (PM) of LSCs and AML cells, such as oxysterol-binding proteins (ORPs), TRPM2, and neurokinin-1 receptor (NK-1R).

ORPs govern phosphatidylinositol-4,5-bisphosphate (PIP2) and cholesterol trafficking to the PM. Zhong et al. reported that abnormally increased expression of ORP4L is crucial for leukemia stem cell survival. It allows inositol-1,4,5-trisphosphate (IP3) formation by removing PIP2 from the plasma membrane and presenting it to phospholipase Cb3 (PLCb3) for hydrolysis. IP3 generation activates IP3Rs, leading to ER Ca^2+^ release to enhance mitochondrial respiration. They described the synthesis of LYZ-81, a molecule that binds ORP4L competitively with PIP2 and inhibits PIP2 hydrolysis, generating a defective Ca^2+^ signaling [68]. The TRPM2 ion channel is activated by free intracellular ADP-ribose in synergy with free intracellular calcium [69].

NK-1R is the high-affinity receptor for substance P (SP). Dysregulation of the SP/NK-1R complex plays a part in multiple pathologies, including pain, chronic inflammation, affective and addictive disorders, and cancer [70,71]. Here, Ge et al. demonstrated in vitro and in vivo the ability of NK-1R antagonists (SR140333 and aprepitant) to induce AML cell apoptosis through IP3R-mediated calcium ER–mitochondrial efflux. Subsequent VDAC-elicited calcium overload induces ROS mitochondrial dysfunction, activating a DNA damage program through ATM and CHK2 activation and resulting in apoptosis [72]. Taken together, these data suggest that blocking Ca^2+^ signaling at the plasma membrane level may combat LSCs and AML cells by targeting their mitochondrial bioenergetic processes.

In addition, several components of the calcium signaling family at the plasma membrane have been described in AML. At the level of purinergic P2 receptors, decreased expression of P2XR3 genes involved in apoptosis mechanisms was observed in EVI-1 AMLs [73]. Conversely, P2X7R upregulates Pbx3, therefore promoting the progression of MLL-AF9 AML [74].

### 3.3. Mitochondrial Calcium and Its Implication in Cancer Mechanisms

Calcium plays a role in many mitochondrial mechanisms in cancer cells. mTORC2–AKT induces direct phosphorylation of IP3R, which induces an escape from cell death by stopping the flow of calcium between the ER and the mitochondria. In liver, lung, breast, and colorectal cancers, GTPase mitofusin-2 (Mfn2) located at the outer membrane of the mitochondria is able to restore cell death by inhibiting the mTORC2–AKT axis [75]. Other proteins can modulate the death pathway of cancer cells, such as Bcl-2, which exerts calcium-dependent oncogenic activity by inhibiting apoptosis stimuli through direct interaction with IP3R and Bax [76]. During apoptosis, the oncosuppressive protein p53 induces the release of calcium from the reticulum to the mitochondria by increasing the activity of the SERCA pump. Likewise, the PML protein can interact with IP3R3 to induce calcium flux from the ER to the mitochondria, thereby affecting apoptosis and autophagy [6].

As mentioned previously, the IP3R receptor located on the ER promotes mitochondrial oxidative phosphorylation by allowing the supply of calcium to the mitochondria. Calcium-dependent apoptosis is predominantly mediated by the IP3R3 variant in mammals. Kuchay et al. argued that the F-box protein FBXL2 is able to bind to IP3R3 to degrade ubiquitin and thus reduce mitochondrial calcium influx. Interestingly, a mutation of PTEN (homologous gene of phosphatase and tensin) is often observed in cancers, in which PTEN is able to compete with FBXL2 by binding to IP3R3, inhibiting cell death by hindering mitochondrial calcium overload [77].

Autophagy is also a mechanism used by cancer cells. The increased expression of VGCC reduces autophagy. Conversely, the decrease in ORAI1 activates autophagy. Additionally, an increase in cytosolic calcium allows the activation of CAMKII, which in turn activates autophagy by regulating the AMPK–mTOR axis [78].

### 3.4. Isocitrate Dehydrogenase in AML

Isocitrate dehydrogenase (IDH) is a crucial cellular enzyme in the TCA cycle. Its main role is to promote the oxidative decarboxylation of isocitrate into α-ketoglutarate. Among the five IDHs identified, three were located in the mitochondrial matrix and were NAD-dependent. The other two are NADP-dependent isocitrate dehydrogenases, one of which is predominantly cytosolic (IDH1) and the other mitochondrial (IDH2). The two NADP-dependent isozymes act as homodimers, and NADP and Ca^2+^ bind in the active site to generate different structures. Xu et al. revealed a new self-regulatory mechanism of activity in which NADP is already binding to the open, inactive form, while the competitive binding of isocitrate and calcium allows conformational changes, resulting in the closed, active form (Figure 2) [79].

Mutations in the IDH1/2 family were reported in metastatic colon cancer in 2006 and then in primary and secondary gliomas two years later following large-scale sequencing [81,82,83]. IDH1/2 mutations were later identified in approximately 20% of patients with AML through whole genome sequencing for IDH1 and by the identification of somatic IDH2 mutations [84,85,86,87]. The frequencies of IDH1 and IDH2 mutations are similar in AML patients and are reciprocally exclusive [88]. The protein structure of mutated IDH1 and IDH2 proteins showed that these enzymes were not inactive but acquired a new active site and the capacity to convert isocitrate to 2-hydroxyglutarate (2-HG) [87,89]. 2-HG was later shown to inhibit hypoxia-inducible factor (HIF1)-α degradation and alter the epigenetic landscape, suggesting that it may function as an oncometabolite. The precise roles of calcium and NADP in the activity of mutated IDH1/2 in AML cells are not known. Computational studies using an allosteric inhibitor suggest that it binds tightly with the divalent calcium cation at the homodimer interface. It then inhibits the formation of the IDH2/R140Q homodimer to a closed conformation that is required for catalysis, resulting in a decrease in NADPH binding [90]. Interestingly, ivosidenib and enasidenib, which are inhibitors of mutated IDH1 and IDH2, respectively, have recently entered phase 1 trial and displayed promising one-year survival effects [91].

In conclusion, there is a striking difference between HSCs and LSCs regarding mitochondrial metabolism, the former relying on both OXPHOS and glycolysis while glycolysis-defective LSCs exploit amino acids, and to a lesser extent fatty acids, to sustain OXPHOS. In addition, LSCs present an abnormal expression of plasma membrane and cytoplasmic proteins that transport calcium to the mitochondria via the endoplasmic reticulum. Finally, mitochondrial enzymes such as IDH are mutated, perturb the Krebs cycle, and produce potential oncometabolites. Thus, the precise role of mitochondrial calcium signaling should deserve closer consideration in order to discover new therapeutic molecules (Figure 3).

## 4. Calcium, Microenvironment, and AML Cells

HSCs reside in close proximity and interact with various cellular components in the BM, such as mesenchymal stem cells, endothelial cells, osteoblasts, osteoclasts, macrophages, and immune cells, including T lymphocytes and natural killers. The term niche is used when nonhematopoietic cells interact with HSCs to influence their functions such as adhesion, quiescence, differentiation, and proliferation by producing cytokines, chemokines, and other soluble factors [93].

Cancer cell behavior still represents a wide research field with many unknowns, such as, for instance, cancer cell dormancy. This is a stage where the tumor microenvironment sends extrinsic signals to suppress cancer cell growth and proliferation before more favorable conditions appear. As conventional therapies target dividing cells, dormant cancer cells that may disperse early during the disease become resistant and thrive as MRD. These cancer cells can then awaken from their dormant state to trigger disease relapse, even long after the treatment ends [46].

AML is characterized by the presence of LSCs, a subpopulation residing in the BM microenvironment and more specifically within niches, where they interact with other types of cells to resist chemotherapy and survive as MRD. As LSCs are dependent on their microenvironments, it is assumed that a better understanding of these microenvironments and how to target them could help reduce MRD and increase patients’ life expectancy [94]. The role of calcium in CSC signaling is still poorly understood, especially in AML, despite its long-known roles in numerous signaling pathways and in the interaction between LSCs and the microenvironment. Here, we describe some of the most important implications of calcium in terms of interactions between LSCs and their microenvironment.

### 4.1. Role of the Endosteal Niche

Bone is the major calcium stock in the body and a key regulatory organ for calcium homeostasis. Bone marrow (BM) consists of a complex hypoxic microenvironment that includes mainly osteoblasts and osteoclasts. Together, they secrete calcium and synthetize the bone matrix, thus forming an interface between calcified bone and marrow cells: the endosteal niche. It has been established that HSCs live in niches within the BM microenvironment that control HSC dormancy, self-renewal, and mobilization through the production of several factors [95,96].

Among those factors, only a few have yet been linked to calcium in the context of AML cancer cell–microenvironment interaction. CXC chemokine 12 ligand (CXCL12, also known as stromal cell-derived factor (SDF-1)) is highly expressed in bone by both mesenchymal stromal cells and osteoblasts. Leukemic stem cells expressing CXC chemokine receptor type 4 (CXCR4) detect chemokine gradients, and, by activating CXCR4, CXCL12 generates an increase in intracellular calcium ion levels. Consequently, the rise in intracellular calcium concentration triggers a chemotactic response, facilitating the entrance of LSCs expressing CXCR4 into the bone microenvironment and thus the niche [97,98,99]. The binding of CXCL12 and CXCR4 activates prosurvival and proliferative signaling pathways, including the PI3K/Akt and MEK/ERK pathways. In contrast, interesting studies have shown that CXCR4 inhibition abolishes the interactions with CXCL12, thus inducing prosurvival signaling. CXCR4 inhibition also prevents LSC anchorage to the BM microenvironment and promotes mobilization of LSCs out of the endosteal niche, thereby rendering them vulnerable to chemotherapy and apoptosis [100,101,102]. These observations led to clinical studies such as the one on the drug Plexirafor, a CXCR4 antagonist, combined with chemotherapy in patients with relapsed or refractory AML [103]. Another feature of the endosteal niche is that physiological bone remodeling through the release of different molecules is also responsible for the high extracellular Ca^2+^ concentration. It can reach 40 mM and appears to be the major cause of normal HSC localization and adhesion in the endosteal niche [104,105].

Some studies have shown that extracellular Ca^2+^ is able to bind to calcium-sensing receptors (CaRs), G-protein-coupled receptors responsible for the regulation of extracellular calcium homeostasis in HSCs residing in the BM [95]. As LSCs also express CaRs, they are sensitive to calcemia, which is controlled by calciotropic hormones such as the parathyroid hormone (PTH) and vitamin D [106]. The binding of extracellular calcium to LSC CaRs stimulates the secretion of PTH-related peptide (PTHrP) through L-type voltage-sensitive calcium channel activation and promotes tumor cell proliferation and survival. CaR deficiency reduces the marrow cellularity and disrupts the localization of LSCs, demonstrating the role of CaRs and PTH as key regulators of the endosteal niche and making them potential targets to reduce LSCs [96,107,108]. Although it has been studied in normal HSCs and CML, for now, we can only assume that this mechanism is common to all leukemic stem cell types, including the AML stem cells residing in the BM for which no data currently exist. Moreover, the precise calcium signaling pathway associated with CaR activation remains elusive, as no calcium channel has explicitly been identified in this context.

### 4.2. Modulation by Retinoic Acid (RA)

As mentioned above, osteoblasts and osteoclasts play a major role in calcium concentration and therefore in AML cancer cells. RA, present in the BM stroma, is also involved in cell growth, proliferation, differentiation, apoptosis, and immune response. RA interacts with its receptors, retinoic acid receptors (RARs) and retinoic acid X receptors (RXRs). They regulate target gene expression in multiple cell types, including AML cells, since RA is abundant in the BM and regulates hematopoietic stem cell renewal [107,108,109]. It has been acknowledged that the action of Ca^2+^/calmodulin-dependent protein kinases (CaMKs) is generated by the binding of Ca^2+^ to calmodulin (CaM) and that Ca^2+^/CaM levels are controlled by the variation of intracellular Ca^2+^ concentration. CaMKs regulate the development and activity of numerous cell types, e.g., cytokine expression in T lymphocytes [110,111].

An interesting study showed a direct interaction between CaMKIIγ and RARs mediated through a CaMKII LxxLL signature pattern in myeloid cells. CaMKIIγ binds RAR target sites within myeloid gene promoters and phosphorylates RAR to inhibit its transcriptional activity, thus regulating myelopoiesis. Inhibition of CaMKII considerably enhances the granulocytic differentiation of acute promyelocytic leukemia cells, meaning that RA prevents LSC differentiation through calcium signaling [112]. Another study proposed that RA-treated leukemic cells display an increase in the expression of the adhesion molecules VLA4, LFA1, VCAM-1, and ICAM1, resulting in an important rise in leukemic cell adhesion to the niche [113]. Similarly, ATRA, a vitamin A derivative, has been shown to have a high affinity for RAR and to induce cell differentiation by increasing the expression of the Ca^2+^ signaling pathway effectors PKC, MAPK, and PI3K, which are required for the activation of transduction pathways leading to cell differentiation [114,115,116,117]. Moreover, ATRA also seems to enhance the production of TGFβ2, which is abundant in the BM and induces cell dormancy through a p38-dependent signaling pathway leading to the activation of the dormancy-associated proteins DEC2/Sharp1 and p27kip1 [107,118].

In summary, these studies show that RA and ATRA regulate LSCs in a calcium-dependent manner, supporting their adhesion to the niche and acquisition of the dormant state, therefore leading to their increased resistance to the immune system and chemotherapy. Even though the Ca^2+^ signaling pathway is still incompletely characterized, these studies have helped develop new chemotherapeutic treatments currently undergoing phase III clinical trials using ATRA to target AML cells [119,120].

### 4.3. The Vascular Niche

Leukemic stem cells interact not only with the endosteal niche but also with the vascular niche. Indeed, HSC activity seems to be linked to vascular development, and this niche contains a large amount of soluble factors and cellular elements that may contribute to leukemia homeostasis [94,121]. Although the vascular niche is being studied in murine models, there are still many gaps in our knowledge of the human vascular niche. What has been shown until now is that mesenchymal stem cells (MSCs) are essential for the well-being of the HSC niche, as they facilitate stem cell engraftment to the vascular niche. MSCs go to the peripheral blood in higher numbers under hypoxic conditions, suggesting that several cell types in the BM niche are also sensitive to hypoxia. LSCs, characterized by a low division rate, come from one of the most hypoxic regions, where they are “hidden” from immune cells and avoid exposure to chemotherapeutic drugs, therefore increasing their survival rate [122].

While it has been documented that a hypoxic microenvironment can have prosurvival effects on AML cells [123,124], studies on the link between hypoxia and calcium remain scarce. Indeed, hypoxia activates the translation of HIF-1α and 2α through an influx of extracellular calcium, the stimulation of PKC, and the activation of mTOR. HIF-1α also directly upregulates the expression of TGFβ1, increasing the expression of CXCR4 on blasts and allowing them to adhere to the niche. As hypoxia limits the recruitment of immune cells and allows leukemic cells protection, drugs such as TH-302 have been developed and used in clinical trials to sensitize formerly resistant leukemic cells to cytarabine, thus inducing apoptosis. The use of TH-302 also appears to decrease, among other things, the expression of HIF-1α [125,126]. In addition, some studies have shown that the inhibition of HIF-2α in primary AML cells seems to inhibit their proliferation [100,124,127,128]. Furthermore, several interesting studies showed that the endothelium and cells located near endothelial cells in the vascular niche called CXCL12-abundant reticular cells (CARs) secrete CXCL12, allowing HSCs, and thus LSCs, to adhere to the vascular niche via the CXCR4 receptor.

The secretion of adhesion factors by a variety of cell types and via different Ca^2+^-dependent mechanisms in microenvironment niches shows the interconnection between those niches and their importance for AML stem cell proliferation and survival. What remains to be identified are the calcium channels and the associated calcium signaling pathways involved in all the mechanisms described above (Figure 4) [97,129].

### 4.4. Immune Escape

Finally, the BM microenvironment also includes immune system cells that are able to control cancer growth through the secretion of soluble factors (e.g., cytokines) or by identifying and killing cancer cells through cytotoxic mechanisms. Tumor dormancy and cancer cell escape from the immune response represent the major causes of relapse in AML, although the underlying mechanisms are still poorly understood [130].

The binding of the target cell major histocompatibility complex (MHC) or antigen to the T lymphocyte T cell receptor (TCR) complex induces the release of ER calcium stores. In turn, ER Ca^2+^ store depletion leads to the oligomerization of the calcium-sensing stromal interaction molecule 1 (Stim1) protein. Then, Stim1 oligomers translocate to regions near the plasma membrane, where they directly bind to calcium-release-activated channels (CRACs) via their STIM1 Orai-activating region (SOAR) domains. This interaction leads to the CRACs opening and subsequent capacitative Ca^2+^ entry, inducing the formation of the Ca^2+^/CaM/calcineurin complex. This calcium influx enables the mobilization of various signaling pathways involving the activation of transcription factors such as NFAT and AP-1 and leads to T cell activation (proliferation, cytokine secretion) or exhaustion. T lymphocyte activation or exhaustion depends on which transcription factors are activated or paired together. For instance, the Ca^2+^/CaM/calcineurin complex can dephosphorylate NFAT, which, within minutes, translocates to the nucleus, where it binds to regulatory sequences and modulates the expression of several genes, including PD-1 and the cytokine IFNγ [127,128,131,132]. An interesting study showed that when activated alone or paired with AP-1, the transcription factor NFAT allows the activation of T lymphocytes. However, another study showed that when paired with the transcription factors BATF or IRF4, NFAT leads to an exhausted phenotype, where T lymphocytes lose their ability to fight cancer cells, synthesize cytokines, and proliferate [133,134].

The mechanisms by which NFAT can lead to either an activated or exhausted phenotype in T lymphocytes are still not entirely known. Nonetheless, interesting studies have helped to establish the link between NFAT and calcium signaling in the expression of PD-1, a molecule expressed on the T lymphocyte surface, and have shown that NFAT could also be associated with PD-L1 expression in targeted cancer cells [135,136].

When a link between PD-1 and its ligand PD-L1, expressed by regulatory cells such as dendritic cells, is established, PD-1/PD-L1 signaling contributes to the modulation of the effector functions of cytotoxic CD8 T cells. Cancer cells can express regulatory ligands such as PD-L1, CD80/CD86, or Galectin-9 to provide a negative regulatory signal to T lymphocytes expressing their respective receptors PD-1, CTLA-4, and TIM-3, resulting in T cell exhaustion [137,138,139].

PD-L1 is expressed in approximately 30% to 60% of leukemic blasts, depending on the patient. An increase in PD-L1 expression in AML patients after MRD has already been shown; it has also been shown that PD-L1 expression is the reason why LSC can resist CD8 T lymphocyte attack after chemotherapy treatment [137,140]. Other important studies also linked the induction of PD-L1 expression by the cytokine IFN-γ secreted by T lymphocytes with AML blast calcium signaling. Indeed, the IFN receptor expressed by tumor cells can activate the Ca^2+^/calmodulin/calcineurin pathway through the production of IP3, leading to the activation of NFAT, which can in turn activate the PD-L1 gene. As a result, AML cells expressing PD-L1 can inhibit T cells presenting the surface molecule PD-1 and prevent them from proliferating and secreting cytotoxic cytokines [141,142]. Additionally, calcium influx activates NFAT, which in turn activates the IFNγ gene promoter, therefore inducing IFNγ production. CD8 T lymphocytes represent an early source of NFAT-dependent IFNγ production during the adaptive response [143]. Studies have shown that in AML, anthracyclines are likely to activate immunogenic cell death (ICD) through cytotoxic T lymphocytes (CTLs) and the secretion of IFNγ. As said before, this secretion of IFNγ allows T cell proliferation as well as an increased expression of PD-L1 on AML cells, and for now, a possible approach to preventing this adaptive resistance and diminishing the risk of relapse could be to associate immune checkpoint inhibition with chemotherapy [144,145].

Thus, the NFAT/Ca^2+^/CDK4 pathway and, more generally, calcium signaling have an impact on the interaction between immune cells and AML cancer cells, especially through the modulation of PD-1/PD-L1 signaling, consequently regulating cytotoxic lysis and CSC survival (Figure 5) [146,147]. Even though a link between calcium, NFAT, and AML-cell-related T lymphocyte exhaustion has been discovered, further studies are required to understand why cancer cells express PD-L1 at different levels and to better characterize the other T lymphocyte inhibitory pathways mentioned above, such as the CTLA-4/CD80/CD86 and TIM-3/Galectin-9 pathways, whose actions in antitumor activity could also be enhanced by finding an upstream target to inhibit [148].

In summary, the BM microenvironment is a regulator of AML cells that influences their functions and changes the course of the disease. Study of the direct interaction between the microenvironment and AML cells has provided growing evidence that (i) calcium signals from the BM microenvironment can release or activate AML cells and (ii) calcium signals from leukemic cells can remodel existing niches to maintain their proliferation and resistance. These findings and the enhanced sensitivity to cytotoxic chemotherapy with related targeting agents suggest that such signals may represent candidate targets for novel therapeutic strategies [121,149,150,151,152].

## 5. Calcium Signaling in AML Treatment: A New Hope?

Despite remarkable research progress, AML remains a hard-to-cure disease. Indeed, in a recent publication summarizing 13 years of clinical trials, Oliva et al. showed that AML treatments still result in relapse for a substantial proportion of patients. Specifically, 46.8% of patients treated with induction chemotherapy exhibited relapse, and the rate decreased to 29.4% when patients were treated by stem cell transplants. While calcium signaling is a well-known contributor to the hallmarks of cancer [153], few clinical trials have focused on pertinent calcium targets. Indeed, most studies have described the impact of drugs already used for other diseases, e.g., VGCC inhibitors for heart-related diseases or hypertension, on AML outcome. This is perfectly illustrated by Chae et al., who presented a 12-year retrospective study on the effect of calcium channel inhibitors [154], where the only reported effect is worse overall survival for patients treated with amlodipine or diltiazem (L-type calcium channel inhibitors). These observations show that dedicated studies are still required to identify the specific calcium channels and associated signaling pathways involved in AML prior to the development of any efficient treatment specifically targeting these pathways.

### 5.1. Chemotherapies, Calcium, and Mitochondria

In the last few years, some studies have started providing this initial background by focusing on the impact of calcium homeostasis on AML progression. Such an example is provided by Chen et al. [25], who showed that the calcium-permeable TRPM2 channel is overexpressed in AML patient cells and in AML cell lines compared to normal precursors. In U937 cells with TRPM2 knockout (KO), they observed a decrease in proliferation associated with a significant decrease in mitochondrial function, namely a decrease in the oxygen consumption rates and ATP production and an increase in reactive oxygen species (ROS) levels. These effects were accompanied by decreases in mitochondrial membrane potential and mitochondrial calcium uptake, thus indicating a profound modification of calcium homeostasis depending on the TRPM2 expression level. Interestingly, the authors observed that TRPM2 KO cells were more sensitive to the chemotherapeutic agent doxorubicin, which induced a strong increase in ROS production. In this model, this effect of TRPM2 KO was linked to the impairment of autophagy through the modulation of the expression of CREB and ATF4 transcription factors. Overall, this study shows that TRPM2 could be an interesting target for AML treatment. However, the main issue is the current absence of specific inhibitors for TRPM2, although a recent publication proposed the A23 compound as a promising new lead for the development of future clinically relevant TRPM2 inhibitors [155].

Several studies have shown that even without specifically targeting TRPM2, mitochondrial activity is altered by potential therapeutic drugs used for the treatment of AML patients. Such an example was recently provided by Wang et al. [156], who developed a new drug, AKI604, that specifically inhibits Aurora kinase A (AURKA). Aurora kinases are known to be overexpressed in several cancers, including AML, where they participate in mitosis and cytokinesis. Aurora kinase inhibitors were thus developed, and one of them (AZD1152) was used in clinical studies as a potential treatment for AML patients after it was shown to decrease AML cell viability and proliferation and to induce apoptosis [157,158]. In their study, Wang et al. showed that AKI604 can revert the effect of signal transducer and activator of transcription 5 (STAT5) on leukemia cells, namely increased proliferation. This result is particularly relevant since STAT5 is known to be aberrantly activated in the blasts of AML patients [159], to lead to decreased sensitivity to tyrosine kinase inhibitors (TKIs) [160], and to control AURKA expression [161]. AKI604 treatment was associated with mitochondrial activity impairment, disruption of mitochondrial membrane potential, and an increase in ROS production. These effects were also associated with an increase in cytoplasmic calcium concentration ([Ca^2+^]_c_), but the study unfortunately did not provide further clues as to the origin and consequences of this modification of calcium homeostasis. However, treatment with AKI604 decreased tumor growth in a xenograft model, thus proving its potential as a therapeutic drug. Another recent study proposed a combination of three drugs to improve AML patient survival, one of which (pimozide) is known as an inhibitor of voltage-gated calcium channels. While it was shown that the combination of the BH3 mimetic ABT-263, mTOR inhibitor AZD 8055, and pimozide was efficient in inducing cell death in resistant AML cell lines and that this effect involved ROS production and the disruption of mitochondrial activity, no evidence was provided regarding the precise impact of calcium homeostasis on this synergistic effect [162].

### 5.2. Modulation of ER Calcium Stores

As presented in this review, several reports have shown a link between potential chemotherapeutic drugs and mitochondrial activity. Drugs can, however, target other key players of the calcium signaling pathway, including inositol 1,4,5 trisphosphate receptors (IP3Rs), whose activation will result in the release of calcium by the ER, an increase in [Ca^2+^]_c_, and the stimulation of numerous calcium signaling pathways. One of those drugs is wogonoside, a flavonoid of natural origin, which was shown to have antiproliferative effects on AML cells via the upregulation of phospholipid scramblase 1 (PLSCR1) [163]. This initial study showed that wogonoside promotes PLSCR1 translocation into the nucleus, where it binds the promoter region of IP3R1 and increases its expression. In a subsequent work, this team tried to better understand the role of calcium homeostasis in the observed effects of wogonoside. Using primary AML cells, they identified several targets modulated by wogonoside and related to its effects on cell proliferation and differentiation; specifically, they found that cyclin-dependent kinase inhibitor 1 (p21Cip1) and 1B (p27Kip1) were upregulated and c-Myc was downregulated. Moreover, they closely monitored the impact of wogonoside on the cytosolic calcium concentration and showed that the drug increased [Ca^2+^]_c_ over time, reaching a peak at 72 h of treatment. Using 2-APB, a broad-spectrum calcium channel inhibitor also targeting IP3Rs, in combination with extracellular calcium removal, they concluded that this increase in [Ca^2+^]_c_ leading to AML cell differentiation was mostly due to calcium release from the ER via IP3R1 and not to calcium influx through the plasma membrane [37]. Interestingly, IP3R2 was also reported by another team to be overexpressed in cytogenetically normal AML and to represent a predictive biomarker associated with a worse prognosis and decreased overall survival [27]. These apparently contradictory observations illustrate the need to further investigate the roles of all IP3R isoforms in AML and of the associated calcium signaling pathways to better understand disease progression and the resistance of this cancer to current chemotherapies.

### 5.3. Chemotherapies Impacting Calcium Influx

Some drugs proposed for the treatment of AML were described as being able to increase [Ca^2+^]_c_ via their direct effect on plasma membrane receptors or calcium channels. Such an example is 4-aminopyridine (4-AP), a commonly used voltage-gated potassium channel inhibitor. The application of 4-AP to AML cell lines was shown to inhibit these channels and to induce apoptosis. However, upon further investigation, Wang et al. demonstrated that the proapoptotic effect of 4-AP was mostly mediated by the activation of the ATP-gated P_2_X_7_ receptor. Indeed, 4-AP application induces calcium entry through this ionotropic receptor located in the plasma membrane, leading to a [Ca^2+^]_c_ increase and to the induction of apoptosis, an effect completely abrogated in AML cells silenced for the P_2_X_7_ receptor [164]. A similar work led to the identification of nutraceutical glucopsychosine, a lipid derived from bovine milk, as a potential antileukemia compound. Glucopsychosine was shown to selectively induce apoptosis in a caspase-independent manner in AML cells, but not in normal hematopoietic cells, as a result of calpain activation. Calpain was activated here by an increase in [Ca^2+^]_c_ resulting from calcium entry through unidentified plasma membrane calcium channels [165]. Another study showed that the farnesyltransferase inhibitor tipifarnib can also induce apoptosis when applied to AML cells by increasing [Ca^2+^]_c_ without directly disrupting ER- or mitochondria-associated calcium signaling. This increase in [Ca^2+^]_c_ was directly linked to the activation of specific plasma membrane calcium channels, namely SOCs, which represent the main calcium entry pathway in non-excitable cells [166]. The pharmacological tools and mRNA screenings used in this study led the authors to propose Orai3 as the main channel involved in the effects of tipifarnib. However, a later study proposed Orai2 and Orai1 as the main components of SOC channels in AML cells, again illustrating the need to better characterize the main calcium entry pathways in this model [167].

In some instances, calcium modulation can result from the ectopic expression of proteins in AML. Such an example is the expression of the olfactory receptor OR51B5 in AML cells, resulting in an increase in [Ca^2+^]_c_ and an inhibition of cell proliferation potentially involving T-type and L-type calcium channels [168]. If confirmed, these results could suggest new targets for innovative therapies targeting either this receptor or the associated signaling pathways.

### 5.4. Future Directions

One of the main signaling pathways activated by extracellular calcium entry or increased intracellular calcium concentrations is the calmodulin/calcineurin/NFAT pathway (for review, see [169]). As shown above, many drugs proposed to treat AML patients induce major variations in [Ca^2+^]_c_, which in turn should dramatically impact the calmodulin/calcineurin/NFAT pathway (Table 2). A recent study by He et al. showed that chemotherapeutic drugs can also directly impact this signaling pathway. Indeed, lenalidomide, a drug used to treat multiple myeloma but with poor reported efficacy in AML, exhibits increased cytotoxic activity in AML when combined with cyclosporine, a well-known inhibitor of calcineurin [170]. This result could therefore suggest that a combination of treatments including modulators of the calmodulin/calcineurin/NFAT pathway could represent a potential way to improve the efficacy of the chemotherapies currently used to treat AML patients. While appealing, this hypothesis remains to be confirmed with the other drugs already known to modulate AML calcium signaling.

Another possible direction for future research is the use of drugs targeting calcium channels expressed by both AML and the tumor microenvironment cells. Indeed, as presented earlier, cells from the different niches directly modulate AML cells’ fate and can promote its progression toward more aggressive stages and survival of chemotherapeutic treatments. In a recently published work, Borella et al. have shown that lercanidipine, a CaV1.2 calcium channel inhibitor, can decrease both AML cell and mesenchymal stromal cell proliferation. Interestingly, they also present evidence that the combination of this dual targeting agent with the chemotherapeutic agent Ara-C significantly decreases tumor growth in a preclinical model, as well as that this effect is far more robust than when each molecule is applied separately [171].

Altogether, these studies highlight the critical role of calcium signaling in AML and the tremendous potential of a better understanding of these pathways when designing the next generation of therapeutic drugs targeting not only AML cells but also their microenvironment.

## Figures and Tables

**Figure 1 cells-11-00543-f001:**
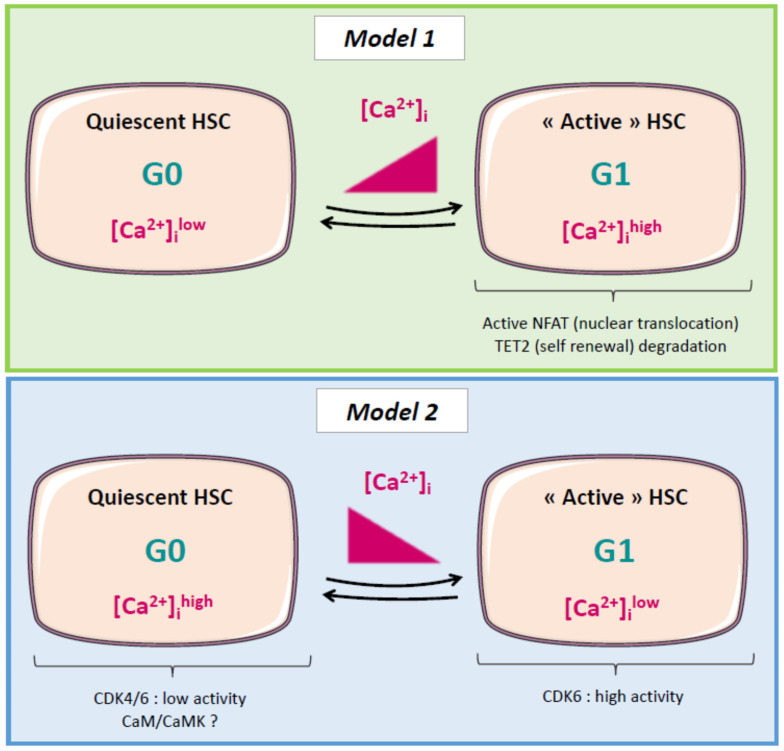
Schematic diagram illustrating the two proposed models of calcium-dependent mechanisms controlling the switch between quiescence and cycling state in HSC. In model 1 [42] and model 2 [9], quiescent hematopoietic stem cells (HSC) display distinct features compared to the cycling/active HSC. In model 1, intracellular Ca^2+^ concentration increases during the switch from G0 to G1 state. Active HSC display active NFAT and TET2 degradation. In model 2, it is proposed that intracellular Ca^2+^ concentration decreases during G0 to G1 transition, leading to an increase in CDK4/6 activities, and that the calmodulin (CaM)/CaM kinase (CaMK) pathway is involved in HSC quiescence.

**Figure 2 cells-11-00543-f002:**
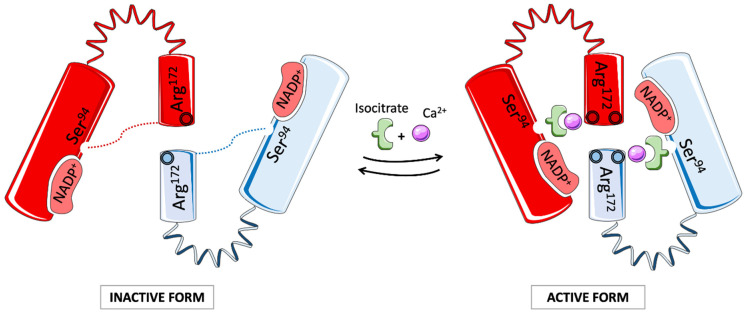
Structure and activation of wild-type IDH2. IDH2 functions as a homodimer, each containing a large NADP+-binding/catalytic domain linked to the small domain by a clasp structure. The open form is maintained by hydrogen bonds between Ser^94^ and Arg^172^, which obstruct the active site. IDH2 becomes activated by switching its conformation. The substrate binding site is made of binding sites for NADP^+^ and isocitrate bound to divalent cations such as Ca^2+^, Mg^2+^, and Mn^2+^. The oxidative catalysis and decarboxylation of isocitrate induce the production of α-KG and NADPH. Mutations in the IDH2 active site provide a gain-of-function activity that converts α-KG to the D2HG oncometabolite (adapted from [79,80]).

**Figure 3 cells-11-00543-f003:**
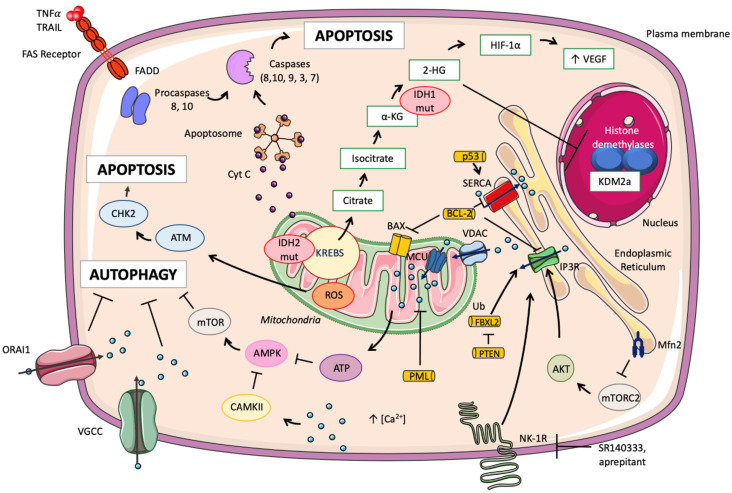
Mitochondrial calcium and its implication in cancer mechanisms. Calcium plays a role in many mitochondrial mechanisms in cancer cells. mTORC2–AKT induces the direct phosphorylation of IP3R, which induces escape from cell death by stopping the flow of calcium between the ER and the mitochondria. In liver, lung, breast, and colorectal cancers, Mfn2 is able to restore cell death by inhibiting the mTORC2–AKT axis [75]. Bcl-2 exerts calcium-dependent oncogenic activity by inhibiting apoptosis stimuli through direct interaction with IP3R and Bax [76]. During apoptosis, the oncosuppressive protein p53 induces the release of calcium from the reticulum to the mitochondria by increasing the activity of the SERCA pump. Likewise, the PML protein can interact with IP3R3 to induce calcium flux from the ER to the mitochondria, thereby affecting apoptosis and autophagy. PTEN is able to block the proteasomal degradation of IP3R3 induced by FBXL2 (F-box protein) [6,77]. Apoptosis can be induced by extra or intracellular pathways. In the extracellular pathway, an extracellular ligand induces the formation of cell death complexes that activate the caspase cascade. In the intrinsic pathway, mitochondrial permeabilization induces the release of cytochrome c, which allows the formation of the apoptosome and then the activation of caspases [6]. In human leukemia cells, apoptosis can occur using inhibitors of NK-1R, which induces calcium efflux from the reticulum to the mitochondria by the IP3R receptor. Because of a lack of space, NK-1R is the only membrane receptor represented in the figure. Mitochondrial dysfunction is induced by the entry of calcium into the mitochondria via VDAC. ROS production and cellular damage induction are induced by the apoptotic axis ATM–CHK2 [72]. Autophagy is also a mechanism adopted by cancer cells; the increased expression of VGCC reduces autophagy, while a decrease in ORAI1 expression activates autophagy. Additionally, an increase in cytosolic calcium allows the activation of CAMKII, which in turn activates autophagy by regulating the AMPK–mTOR axis [78]. In cancer cells, IDH1 is mutated in the cytoplasm, and IDH2 is mutated in the mitochondria. These enzymes allow the production of oncometabolite 2-HG from α-KG, which induces a modification of the methylation of histones [92].

**Figure 4 cells-11-00543-f004:**
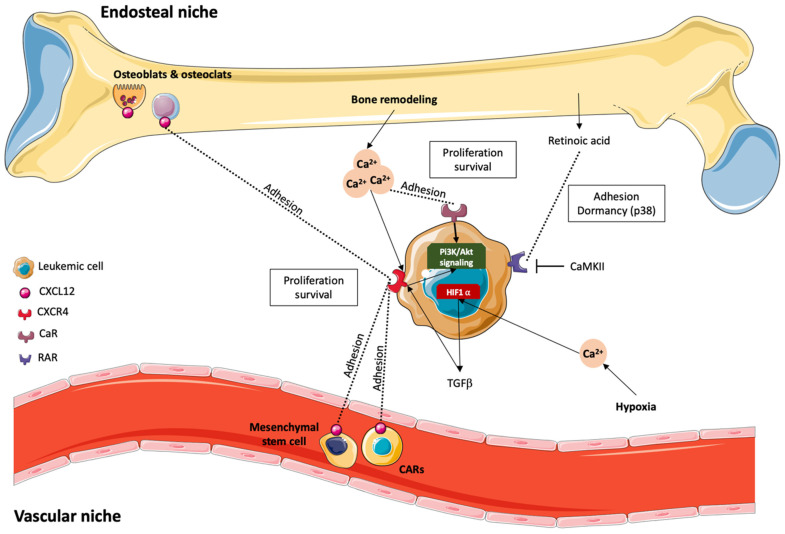
Mechanisms of adhesion to the niche. The binding of CXCL12 expressed by mesenchymal stromal cells, CARs, osteoclasts, and osteoblasts to CXCR4 expressed by leukemic stem cells induces an increase in intracellular calcium ion levels, leading to the entrance of LSCs into the vascular or endosteal niche, which in turn induces prosurvival and proliferative signaling pathway mobilization. Bone remodeling is responsible for the high extracellular calcium level, which enables the binding of calcium to the CaRs expressed by LSCs, thus helping the cells adhere to the niche. RA present in the BM microenvironment can bind to its RARs expressed on LSCs, inducing the regulation of LSC growth, proliferation, differentiation, apoptosis, and immune response; however, RAR activity can be modulated by CaMKII. The hypoxic environment of the BM seems to have proliferation-inducing and prosurvival effects on AML cells and upregulates CXCR4 expression on blasts, allowing them to adhere to the niche.

**Figure 5 cells-11-00543-f005:**
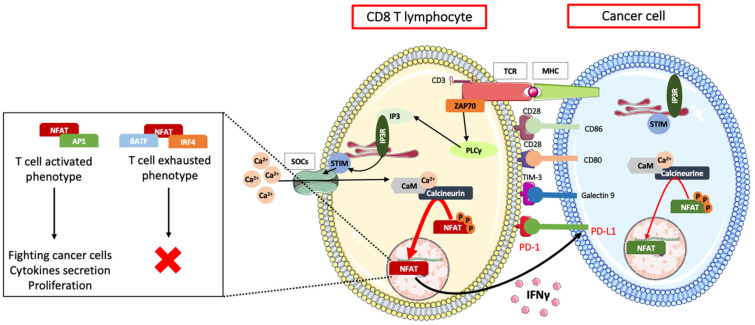
Cancer cell immune escape. The binding of the target cell’s MHC or antigen to the T lymphocyte TCR complex induces the release of ER calcium stores, leading to Stim1 oligomerization and allowing calcium entry through plasma membrane CRAC channels. Capacitive Ca^2+^ entry induces the formation of the Ca^2+^–CaM–calcineurin complex, which in turn dephosphorylates the transcription factor NFAT, which, within minutes, translocates to the nucleus. When activated alone or with the other transcription factor AP1, NFAT allows T lymphocyte activation, and when paired with BATF or IRF4, NFAT can induce an exhausted phenotype. When bound to regulatory sequences, NFAT can modulate PD-1 expression and IFNγ cytokine secretion. When the link between PD-1 expressed by T lymphocytes and its ligand PD-L1 expressed by cancer cells is established, PD-1/PD-L1 signaling contributes to the modulation of effector functions of cytotoxic CD8 T cells. Cancer cells can also express other regulatory ligands, such as CD80/CD86 or Galectin-9, to provide a negative regulatory signal to T lymphocytes expressing their respective receptors CTLA-4 and TIM-3, resulting in T cell exhaustion. IFNγ secreted by T lymphocytes can bind to its receptor expressed by tumor cells and can thus activate the Ca^2+^/CaM/calcineurin pathway through the production of IP3, leading to the activation of NFAT, which can in turn activate the PD-L1 gene, inducing the inhibition of T cells presenting the surface molecule PD-1 and therefore preventing them from proliferating and secreting cytotoxic cytokines. Additionally, calcium influx activates NFAT1, which in turn activates the IFNγ gene promoter, inducing IFNγ production and thus regulating cytotoxic lysis and CSC survival.

**Table 1 cells-11-00543-t001:** Summary of the main targets/pathways of interest associated with calcium signaling and related to AML proliferation and differentiation.

Sub-Sections	Targets	Mechanisms	Biological Effect	Biological Sources	References
**Ca^2+^ Signaling and Cell Cycle Regulation**	CaM	-Increased cytosolic CaM -Transition G1 to S	Increased proliferation	HL60 promyelocytic AML cell line	[17]
	CaMKII	-Decreased Cdk inhibitors p27 (kip1) and p16 (ink4a) -Increased cyclin A, B1, D1	-Increased proliferation -Cell cycle progression	AML cell lines	[19]
	CaMKIV	-Increased p27, p16 -Decreased cyclin A, B1, D1	-Decreased proliferation	AML cell lines	[19]
	CaMKIV	-Phosphorylation Rb	Increased proliferation	Primary AML cells	[18]
	Calcineurin	-Transition G1 to S -Cyclin A, D1? -Cyclin E, E2?	Proliferation? Decreased calcineurin activity (−85%)	Sera from AML patient	[24]
**Ca^2+^ Channels and Proliferation in AML**	TRPM2	ATF4, CREB	Increased proliferation	Primary AML cells	[25]
	Cav.1.2, L-type calcium channel	-Ca^2+^ entry	Increased proliferation	Primary AML cells	[26]
	ITPR2	SERCA pumps	Cell cycle progression	Primary AML cells	[27]
**Notch and Ca^2+^ Signaling in AML Proliferation**	Notch/Delta ligand	-Calcium sensor receptor -SOCE	Increased proliferation	TMD7 AML cell line	[29]
	Notch/Delta1 ligand	-Calcium sensor receptor -SOCE	Increased proliferation	HL60 promyelocytic cell line	[30]
	Notch/Jagged1 ligand	-Calcium sensor receptor -SOCE	Decreased proliferation	Primary AML cells, AML cell lines	[28,33]
**Calcium Involvement in AML Differentiation**	SERCA pumps	-Ca^2+^	Increased differentiation	Primary AML cells	[34]
	S100A9/TLR4	-p38, ERK1/2, JNK	Increased differentiation	Primary AML cells	[35]
	Ca^2+^ concentration		Increased differentiation	Primary AML cells	[36]
	IP3R1	-Decreased c-myc expression	Increased AML cells	AML cell lines	[37]

**Table 2 cells-11-00543-t002:** Summary of the main molecules with chemotherapeutic potential targeting the calcium signaling pathway in AML.

Molecules	Targets	Clinical Use	Clinical Impact	Mechanism	Ref
**Amlodipine/** **Diltiazem**	L-type calcium channels	Yes (heart disease, hypertension)	Decreased AML patient survival	L-type calcium channels inhibitors	[154]
**A23**	TRPM2 channel	No	-	TRPM2 inhibitor makes AML cells more sensitive to chemotherapies in vitro (increases ROS production)	[25]
**AKI604**	Aurora kinase A (AURKA)	No	-	AURKA inhibitor impairs mitochondrial activity, increases ROS production and cytoplasmic calcium concentration, and decreases tumor growth in xenograft models	[156]
**Pimozide**	Voltage-gated calcium channels	No	-	In combination with ABT-263 and AZD 8055, pimozide impairs mitochondrial functions and induce resistant AML cell lines apoptosis	[162]
**Wogonoside**	IP3R1	No	-	Inhibits proliferation through PLSCR1 activation, IP3R1 upregulation, and the resulting increase in cytoplasmic calcium concentration leading to AML cell differentiation	[37,163]
**4-AP**	Voltage-gated potassium channel	No	-	Inhibition of voltage-gated potassium channels by 4-AP leads to plasma membrane. depolarization, calcium entry into AML cells via ionotropic P_2_X_7_ receptor, and induction of apoptosis	[164]
**Glucopsychosine**	Unknown	No	-	Induces apoptosis in AML cells, but not in normal hematopoietic cells, via a calcium entry through unknown calcium channels	[165]
**Tipifarnib**	Farnesyltransferase	No	-	Tipifarnib inhibits farnesyltransferase and increases intracellular calcium concentration through SOC channels activation, leading to AML cell apoptosis	[166]

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
