# Peer review of "Put in a “Ca^2+^ll” to Acute Myeloid Leukemia"

_cells, 2022, doi:10.3390/cells11030543_

Round 1

Reviewer 1 Report

This review systematically summarized present research on the biological and molecular functions of calcium in the determination of AML cell fate including proliferation, differentiation, metabolism, and their contributions to the maintenance of LSCs stemness, tumor microenvironment and immune escape etc. which may be or have been proved to be involved in the AML initiation, progression and relapse. Thus, indicating that targeting calcium related signaling or channels could be a potential therapy strategies on the treatment of AML.

The contents and summary are informative and comprehensive, however, more precise tables and charts can be better. The authors should pay more attention to improve their English writing skills. The following are some suggestions on this review.

1, The contents in Abstract and introduction parts are overlapped.

2, Page 4. “d) Implication of Notch and Ca2+ signaling in AML proliferation”

  Different subtypes of Notch ligands show diverse effects on leukemia progression, especially on AML, thus specific classification and summary need to be clarified.

“Notch activation inhibits AML growth and survival: a potential therapeutic approach” DOI: 10.1084/jem.20121527

Notch pathway activation targets AML-initiating cell homeostasis and differentiationDOI: 10.1084/jem.20121484

3, Page 12 “While insufficiently documented, a hypoxic microenvironment seems to have pro- survival effects on AML cells.”

  Some papers have reported that hypoxic microenvironment have effects on leukemia maintenance and proliferation.

Hypoxia-Activated Prodrug TH-302 Targets Hypoxic Bone Marrow Niches in. Preclinical Leukemia Models” DOI: 10.1158/1078-0432.CCR-14-3378

“Hypoxia-induced upregulation of BMX kinase mediates therapeutic resistance in acute myeloid leukemia” DOI: 10.1172/JCI91893

Author Response

This review systematically summarized present research on the biological and molecular functions of calcium in the determination of AML cell fate including proliferation, differentiation, metabolism, and their contributions to the maintenance of LSCs stemness, tumor microenvironment and immune escape etc. which may be or have been proved to be involved in the AML initiation, progression and relapse. Thus, indicating that targeting calcium related signaling or channels could be a potential therapy strategies on the treatment of AML.

The contents and summary are informative and comprehensive, however, more precise tables and charts can be better. The authors should pay more attention to improve their English writing skills. The following are some suggestions on this review.

We wish to thank our reviewer for his/her careful reading and positive evaluation of our manuscript. Following the reviewer’s recommendations, we have added two tables summarizing the main calcium signaling pathways associated with AML cells proliferation/differentiation (table 1) and chemotherapies (table 2). Prior to the initial submission, our manuscript was edited by a private company (https://www.aje.com/) in order to improve our English. We are therefore surprised by this remark, but will take advantage of the Cells editing team in order to improve the manuscript wherever necessary.

1, The contents in Abstract and introduction parts are overlapped.

 We acknowledge this oversight, and have thus modified both the abstract and the introduction.

2, Page 4. “d) Implication of Notch and Ca2+ signaling in AML proliferation”

  Different subtypes of Notch ligands show diverse effects on leukemia progression, especially on AML, thus specific classification and summary need to be clarified.

 “Notch activation inhibits AML growth and survival: a potential therapeutic approach” DOI: 10.1084/jem.20121527

“Notch pathway activation targets AML-initiating cell homeostasis and differentiation” DOI: 10.1084/jem.20121484

We thank the reviewer for his/her comments. Indeed, according to the subtype of Notch ligands (i.e., Jagged1 or Delta 1), activation of this pathway leads to opposite effects on AML cells proliferation. We have therefore clarified this specific point in the revised version of the manuscript, and we have now included both references pointed out by our reviewer in addition to the one already mentioned in the original version in order to better explain the opposite effects of Notch ligands.

3, Page 12 “While insufficiently documented, a hypoxic microenvironment seems to have pro- survival effects on AML cells.”

  Some papers have reported that hypoxic microenvironment have effects on leukemia maintenance and proliferation.

Hypoxia-Activated Prodrug TH-302 Targets Hypoxic Bone Marrow Niches in. Preclinical Leukemia Models” DOI: 10.1158/1078-0432.CCR-14-3378

“Hypoxia-induced upregulation of BMX kinase mediates therapeutic resistance in acute myeloid leukemia” DOI: 10.1172/JCI91893

We thank the reviewer for his/her corrections and for pointing out extra examples supporting our statements. There are indeed papers mentioning the effect of hypoxia on cancer cells and more specifically on AML cancer cells. We did say “insufficiently documented” as this part of the review focuses on the role of calcium in the tumor microenvironment and, as of today, the only papers we found concerning hypoxia and calcium are those already mentioned. However, taking into account the reviewer’s remark, we have nuanced our idea, and included the references mentioned.

Reviewer 2 Report

Ca2+ is a versatile intracellular signal that regulates every aspect of cellular life. Although recent studies have revealed the important role of Ca2+ signal in hematopoietic stem cells and solid cancers, little is known about the role of Ca2+ in acute myeloid leukemia (AML). This review summarizes the current understanding of calcium signaling in AML biology, including the role of Ca2+ signal in AML stem cells and the role of Ca2+ in microenvironment of AML. The authors also describe Ca2+ targeting therapies for AML.

This review is very comprehensive, well-written and well-organized, providing valuable information in this field. I only have a few comment to this insightful review.

(1) Page 2, page 11, page 14, some letters appear larger than others.

(2) In the title, what does the “Ca2+ll” mean?

Author Response

Ca2+ is a versatile intracellular signal that regulates every aspect of cellular life. Although recent studies have revealed the important role of Ca2+ signal in hematopoietic stem cells and solid cancers, little is known about the role of Ca2+ in acute myeloid leukemia (AML). This review summarizes the current understanding of calcium signaling in AML biology, including the role of Ca2+ signal in AML stem cells and the role of Ca2+ in microenvironment of AML. The authors also describe Ca2+ targeting therapies for AML.

This review is very comprehensive, well-written and well-organized, providing valuable information in this field. I only have a few comment to this insightful review.

We wish to thank our reviewer for his/her careful reading and very positive evaluation of our manuscript.

(1) Page 2, page 11, page 14, some letters appear larger than others.

Indeed, as pointed out by the reviewer, some typographical errors are present in the manuscript. Since these were not present in our original manuscript, we believe they are linked to the editing made by the journal. We will thus make sure to correct them before any potential publication of the manuscript.

(2) In the title, what does the “Ca2+ll” mean?

We placed the symbol for calcium (Ca2+) in the word “Call” as an extra reference to the central role played by calcium in AML, as hopefully shown by this review.

Reviewer 3 Report

Abstract: Very clear summary of the review article: introduction to AML and calcium as well as the focus of the review article and the four main, specific topics that it focuses on.  Clear, concise, and well laid out.

Introduction:  Expanded section based on the abstract section: again, clear explanations and introductions to AML; disease, classification as well as a small introduction to haematopoiesis

  • Could be expanded on, particularly as the review focuses a lot later on the role of HSCs and the bone marrow micro-environment niche.
  • Another suggestion here could be the inclusion of current treatments for AML (chemotherapy intervention - specific drugs used) - again later on the review mentions doxorubicin, this could also be incorporated into this section.
  • An accompanying figure / schematic to summarise its role within the cell and its link to tumorigenesis would be a nice inclusion

I liked the final sections leeway into the focuses of the article and the lack of understanding as to the role of calcium signalling in AML disease, compared to CML (Bcr-Abl). This sets the scene for the rest of the review article nicely. Additionally, the article sets out clearly that it will be focussing on calcium channels as well as downstream intracellular signalling pathways which are consistent themes throughout the article.

Section (2): The role of calcium homeostasis in AML cell proliferation and differentiation.

The opening section lays out nicely the focus of the section - effects of calcium signalling dysregulation. It was good to see the source material for the review clearly outlined AML cell lines and primary samples.

  • However, overall, I found this section compared to the others in the review to be very text and 'science' heavy. There is a lot of complex biology explained across a range of areas; effect of calcium signalling on cell cycle regulators - this section is broken up into different 'targets' and areas but again I found it to be very heavy in text and the biology and at times complex and hard to follow,
    • proliferation in AML - this section was easier to follow and understand,
    • Notch signalling - smaller section and was a nice inclusion to the article, differentiation and effects on normal and cancer stem cells
    • these sections were interesting and included a range of examples of calcium signalling modifying and being linked to these AML features.
    • However, the inclusion of conflicting studies for me reinforces that whilst this review article does cover a lot of areas, section (2) for  was the 'weakest' in terms of providing evidence of links between calcium signalling and AML differentiation and cell cycle progression.
    • There were nice summaries of these processes, however without the inclusion of accompanying diagrams for me this section was very text heavy and quite hard to process and understand.
    • Another possible inclusion could even be a summary table on the seven individual sections within this chapter - what were the main targets / pathways of interest, what were the mechanisms identified (so far), with a possible conclusions sections where the authors views on conflicting studies and the need for future studies on these areas highlighted. 

I would add however, that the science and referencing here is all accurate and fine and with multiple (note multiple required) readings there is a lot of high-quality information; role of calcium signalling in quiescence (CDK4/6 expression and store operating channels) and AML differentiation (IP3R1 and CaMK pathways - key themes throughout the article which could possibly have their own sections?). Smaller sections on AML relapse are agood inclusions - with similar themes of future investigations being needed to fully elucidate some of the mechanisms that the authors highlight (i.e.. over-expression of NFAT isoforms) being a common conclusion and theme throughout the article.

  • Furthermore, another possible inclusion to the article could be a discussion on relevant techniques for studying these processes (RNA-seq is mentioned) but imaging techniques could also be mentioned possibly?

Section (3): Mitochondria, calcium and AML

One of the strongest sections in this review. Clearly laid out in a nice, relevant order - introduction to mitochondria, normal haematopoiesis, AML, implications in cancer mechanisms and IDH (AML).

Really found the graphical / summary abstract helpful for summarising everything discussed in this section. Key themes; IP3R, Bcl-2, PTEN, ORA1L and CaMKII - localisation in the cell / functions / relevance of calcium signalling. 

  • Minor comments: not as convinced by the final two sections, specifically the potential use of BRAF inhibitors (relevance from a different disease model - more information for AML would be nice here), IDH in AML - the functional literature and the relevance to calcium is nice.
  • It may be useful to include clinical and background information on IDH inhibitors for AML treatment - are there any studies that have linked their role and mechanism of action to modulation of calcium signalling and calcium homeostasis in the cell?
  • Are there also any small figures to properly highlight the link between calcium and IDH's transition to its active form structure?

Section (4): Calcium, microenvironment and AML cells

This was a complex section but the inclusion of two summary / graphical abstracts made this section much easier to follow and understand. 

A few examples for the sections were picked out and discussed well: CXCR4, CXCL12 and their roles in activating survival and proliferation pathways (PI3K/Akt and MEK/ERK). I also liked the section of ATRA and retinoic acid and was convinced that they both seem very prominent areas for future research - with ATRA being mentioned as already entering phase III clinical trials. 

  • The roles of calcium signalling in LSC adherence to the niche as well as mTOR activation were also clearly discussed. However, the authors do mention that no calcium channel has been identified to accompany the discussion and mechanistic work on the vascular niche section which makes the section slightly less convincing and could be expanded to include what areas and targets are currently being investigated (at in vitro level) / what needs to be considered for targets to be identified? / how does this fit into the summary graphic and the other targets (proteins) and pathways also highlighted.
  • Finally, the last section which focuses on immune escape and its link with calcium signalling, I found the most interesting and relevant - particularly the discussion on PD-L1 and IFN-y (what could this mean for the use of immunotherpaies that target these checkpoints in AML therapy?). Stim1 protein was highlighted as the key, inital mediator of some of these calcium-relevant processes and its overalm link and effects on T cell function are also discussed clearly and highlighted nicely in the summary abstract.

Overall, a really enjoyable section with the complex niche biology duly explained by the summary graphical abstract figure.

Section (5): Calcium signalling in AML treatment: A new hope?

Firstly, this section of the review article for me is the most important - focusing on the therapeutic potential of targeting and modifying calcium signalling and homeostasis for AML treatment.

The article again is separated clearly into three separate sections, all of which contain different examples of targets for new therapies; TRPM2, AURKA, IP3R isoforms as well as already discovered novel therapies; wogonoside, tipifarnib, 4-AP - as well as relvant information on their mechanisms of action. 

  • Possible inclusions for this section could be smaller diagrams of the targets (TRPM2 and P2X7R) as well as a summary table of all the targets, drugs and their mechanisms of action. I wasn't also convinced by the final example of combination treatments with other chemotherapy drugs (lenalidomide) - for these combination, synergistic treatments more information on their mechanisms of action would have been helpful.

Furthermore, a final future directions / summary discussion section may also have been helpful for the review, given that it covers a large and wide range of areas - all of which are relevant and explained in great detail but can be lost in focus towards the end of the review. 

Author Response

We wish to thank our reviewer for his/her careful reading of our manuscript, the many helpful comments and suggestions present in the report, and the overall very positive evaluation of our work.

Abstract: Very clear summary of the review article: introduction to AML and calcium as well as the focus of the review article and the four main, specific topics that it focuses on.  Clear, concise, and well laid out.

Introduction:  Expanded section based on the abstract section: again, clear explanations and introductions to AML; disease, classification as well as a small introduction to haematopoiesis

  • Could be expanded on, particularly as the review focuses a lot later on the role of HSCs and the bone marrow micro-environment niche.
  • Another suggestion here could be the inclusion of current treatments for AML (chemotherapy intervention - specific drugs used) - again later on the review mentions doxorubicin, this could also be incorporated into this section.
  • An accompanying figure / schematic to summarise its role within the cell and its link to tumorigenesis would be a nice inclusion

As suggested by our reviewer, we have included a new paragraph introducing HSCs and the BM niche in the introduction. We however chose not to add in this section any mention of current treatments, so as to keep our primary focus on calcium. Similarly, we believe that the addition of a figure in this section of the review would be more confusing than helpful to the readers.

I liked the final sections leeway into the focuses of the article and the lack of understanding as to the role of calcium signalling in AML disease, compared to CML (Bcr-Abl). This sets the scene for the rest of the review article nicely. Additionally, the article sets out clearly that it will be focussing on calcium channels as well as downstream intracellular signalling pathways which are consistent themes throughout the article.

Section (2): The role of calcium homeostasis in AML cell proliferation and differentiation.

The opening section lays out nicely the focus of the section - effects of calcium signalling dysregulation. It was good to see the source material for the review clearly outlined AML cell lines and primary samples.

  • However, overall, I found this section compared to the others in the review to be very text and 'science' heavy. There is a lot of complex biology explained across a range of areas; effect of calcium signalling on cell cycle regulators - this section is broken up into different 'targets' and areas but again I found it to be very heavy in text and the biology and at times complex and hard to follow,
    • proliferation in AML - this section was easier to follow and understand,
    • Notch signalling - smaller section and was a nice inclusion to the article, differentiation and effects on normal and cancer stem cells
    • these sections were interesting and included a range of examples of calcium signalling modifying and being linked to these AML features.
    • However, the inclusion of conflicting studies for me reinforces that whilst this review article does cover a lot of areas, section (2) for  was the 'weakest' in terms of providing evidence of links between calcium signalling and AML differentiation and cell cycle progression.
    • There were nice summaries of these processes, however without the inclusion of accompanying diagrams for me this section was very text heavy and quite hard to process and understand.
    • Another possible inclusion could even be a summary table on the seven individual sections within this chapter - what were the main targets / pathways of interest, what were the mechanisms identified (so far), with a possible conclusions sections where the authors views on conflicting studies and the need for future studies on these areas highlighted. 

We would like to thank the reviewer for his/her helpful comments and suggestions. Indeed, we do agree that the section 2 is text and « science » heavy. Therefore, we have significantly modified the text in order to make easier to follow. Parts 2 a) and 2b) have been fused in a single one so as to make our message more understandable, and to better emphasize the known links between calcium signaling and AML cell progression and differentiation.

As suggested by our reviewer, and to further clarify the processes described in this section, we have now included Table 1 which summarizes the main targets / pathways of calcium, the associated mechanisms, and the reported biological effects on AML proliferation and differentiation. Table 1 also mentions the biological material used in these studies (i.e., cell lines or primary cells). Moreover, rather than including a general schematic for this section that could prove confusing, we added new figure 1 in order to illustrate the two proposed models of the « calcium signature » of quiescent HCS vs active HSC.

Following the reviewer suggestion, we have included a final section entitled « Future directions » where we discuss conflicting studies and suggest possible paths for future investigations.

I would add however, that the science and referencing here is all accurate and fine and with multiple (note multiple required) readings there is a lot of high-quality information; role of calcium signalling in quiescence (CDK4/6 expression and store operating channels) and AML differentiation (IP3R1 and CaMK pathways - key themes throughout the article which could possibly have their own sections?). Smaller sections on AML relapse are agood inclusions - with similar themes of future investigations being needed to fully elucidate some of the mechanisms that the authors highlight (i.e.. over-expression of NFAT isoforms) being a common conclusion and theme throughout the article.

  • Furthermore, another possible inclusion to the article could be a discussion on relevant techniques for studying these processes (RNA-seq is mentioned) but imaging techniques could also be mentioned possibly?

While we fully agree that a review on the relevant techniques for studying calcium signaling and cellular processes such as proliferation or differentiation is of great interest, we believe that such an endeavor is beyond the scope of this manuscript, and deserves to be treated in a dedicated manuscript.

Section (3): Mitochondria, calcium and AML

One of the strongest sections in this review. Clearly laid out in a nice, relevant order - introduction to mitochondria, normal haematopoiesis, AML, implications in cancer mechanisms and IDH (AML).

Really found the graphical / summary abstract helpful for summarising everything discussed in this section. Key themes; IP3R, Bcl-2, PTEN, ORA1L and CaMKII - localisation in the cell / functions / relevance of calcium signalling. 

  • Minor comments: not as convinced by the final two sections, specifically the potential use of BRAF inhibitors (relevance from a different disease model - more information for AML would be nice here), IDH in AML - the functional literature and the relevance to calcium is nice.

We thank the reviewer for his/her helpful comments, and we understand his/her concern about the relevance of BRAF in this section. We therefore decided to remove the corresponding sentences from this paragraph, as it indeed helps keeping the manuscript more focused on AML.

  • It may be useful to include clinical and background information on IDH inhibitors for AML treatment - are there any studies that have linked their role and mechanism of action to modulation of calcium signalling and calcium homeostasis in the cell?

We fully agree with this comment, but as far as we can find, there is currently no data available connecting IDH inhibitors to calcium signaling.

  • Are there also any small figures to properly highlight the link between calcium and IDH's transition to its active form structure?

Following the reviewer suggestion, we have now added new figure 2 depicting the effect of calcium on IDH transition from the open inactive form to the closed active form.

Section (4): Calcium, microenvironment and AML cells

This was a complex section but the inclusion of two summary / graphical abstracts made this section much easier to follow and understand. 

A few examples for the sections were picked out and discussed well: CXCR4, CXCL12 and their roles in activating survival and proliferation pathways (PI3K/Akt and MEK/ERK). I also liked the section of ATRA and retinoic acid and was convinced that they both seem very prominent areas for future research - with ATRA being mentioned as already entering phase III clinical trials. 

  • The roles of calcium signalling in LSC adherence to the niche as well as mTOR activation were also clearly discussed. However, the authors do mention that no calcium channel has been identified to accompany the discussion and mechanistic work on the vascular niche section which makes the section slightly less convincing and could be expanded to include what areas and targets are currently being investigated (at in vitro level) / what needs to be considered for targets to be identified? / how does this fit into the summary graphic and the other targets (proteins) and pathways also highlighted.

We agree with the reviewer: the link between the vascular niche and calcium is still poorly understood, but we have unfortunately not been able to find any new study beyond the ones already cited delving deeper into the identification of the calcium signaling pathways or its related targets. Moreover, as this review primarily focuses on calcium, we chose not to go too far on areas and targets unrelated to calcium in order to prevent the risk of being irrelevant.

However, taking into account the reviewer’s concerns, we have expended this paragraph and added references pointing to drugs targeting factors that could be linked to calcium signaling.

We decided to include one reference on a CXCR4 antagonist: Uy GL et al. A phase 1/2 study of chemosensitization with the CXCR4 antagonist plerixafor in relapsed or refractory acute myeloid leukemia. Blood. 2012;119D17]:3917–24; and 3 references on hypoxia-targeting drugs:

Jensen PO et al. Increased cellular hypoxia and reduced proliferation of both normal and leukaemic cells during progression of acute myeloid leukaemia in rats. Cell Prolif. 2000;33(6):381–95.

Portwood S et al. Activity of the hypoxia-activated prodrug, TH-302, in preclinical human acute myeloid leukemia models. Clin Cancer Res. 2013;19D23]:6506–19.

Handisides DR, et al. A Phase 1 Study Of TH-302, An Investigational Hypoxia-Targeted Drug, In Patients With Advanced Leukemias. 2013;122:3920-3920.

  • Finally, the last section which focuses on immune escape and its link with calcium signalling, I found the most interesting and relevant - particularly the discussion on PD-L1 and IFN-y (what could this mean for the use of immunotherpaies that target these checkpoints in AML therapy?). Stim1 protein was highlighted as the key, inital mediator of some of these calcium-relevant processes and its overalm link and effects on T cell function are also discussed clearly and highlighted nicely in the summary abstract.

In order to address the questions raised by the reviewer, we have added a paragraph on immune checkpoint inhibition associated with chemotherapy, and included 2 new references:

  1. Stahl M, Goldberg AD. Immune Checkpoint Inhibitors in Acute Myeloid Leukemia: Novel Combinations and Therapeutic Targets. Curr Oncol Rep. avr 2019;21(4):37.
  2. Sehgal A, Whiteside TL, Boyiadzis M. Programmed death-1 checkpoint blockade in acute myeloid leukemia. Expert Opinion on Biological Therapy. 3 août 2015;15(8):1191‑203.

We also added a sentence about upstream targets with an extra reference:

Segovia M, Russo S, Jeldres M, Mahmoud YD, Perez V, Duhalde M, et al. Targeting TMEM176B Enhances Antitumor Immunity and Augments the Efficacy of Immune Checkpoint Blockers by Unleashing Inflammasome Activation. Cancer Cell. mai 2019;35(5):767-781.e6

Overall, a really enjoyable section with the complex niche biology duly explained by the summary graphical abstract figure.

Section (5): Calcium signalling in AML treatment: A new hope?

Firstly, this section of the review article for me is the most important - focusing on the therapeutic potential of targeting and modifying calcium signalling and homeostasis for AML treatment.

The article again is separated clearly into three separate sections, all of which contain different examples of targets for new therapies; TRPM2, AURKA, IP3R isoforms as well as already discovered novel therapies; wogonoside, tipifarnib, 4-AP - as well as relvant information on their mechanisms of action. 

  • Possible inclusions for this section could be smaller diagrams of the targets (TRPM2 and P2X7R) as well as a summary table of all the targets, drugs and their mechanisms of action. I wasn't also convinced by the final example of combination treatments with other chemotherapy drugs (lenalidomide) - for these combination, synergistic treatments more information on their mechanisms of action would have been helpful.

As suggested by the reviewer, new Table 2 summarizing the main targets and drugs presented in this section has been added to the manuscript. While TRPM2 and P2X7 receptors are indeed important targets, we believe that the addition of figures presenting their structure is beyond the point of this review. Taking into account the reviewer’s comment, and in the current absence of data validating our hypothesis, the conclusion on lenalidomide therapeutic potential has been tuned down.

Furthermore, a final future directions / summary discussion section may also have been helpful for the review, given that it covers a large and wide range of areas - all of which are relevant and explained in great detail but can be lost in focus towards the end of the review. 

We have modified the conclusion for this section, and added a “future directions” paragraph, as suggested by the reviewer. We have also extended this section by including a recent work by Borella et al (new reference 176) showing the potential interest of targeting calcium signaling in both AML cells and their microenvironmen

Round 2

Reviewer 3 Report

My previous comments have been considered by the authors and where appropriate been addressed or where they are not able to address them due to lack of evidence this has been commented on in their rebuttal.

An excellent review worthy of publication